# Discretization and Predictive Complexity of Time Series

## Abstract

Discretization is widely used in time-series analysis to convert continuous observations into symbolic sequences before sequence modeling. Its effect on forecasting, however, is not merely representational: discretization may preserve the predictive structure of the original process, or it may destroy it by merging histories that imply different future distributions. In this paper, we study discretization through the lens of predictive states and Hankel-rank-based predictive complexity. We first formalize predictive-sufficient discretization and review how predictive-state collapse under coarsening reduces predictive complexity. We then introduce synthetic same-$K$ hidden Markov model families that share the same hidden-state cardinality but exhibit different Bayes-level context gaps. These families allow us to separate nominal hidden-state size from observable predictive difficulty in a controlled setting. Our experiments show that hidden-state count alone does not determine forecasting difficulty, even when the latent-state cardinality is fixed. Moreover, learner-side recovery of Bayes-level context sensitivity is family-dependent and non-monotone in hidden dimension: some families benefit from moderate increases in representation size, whereas others degrade when the model dimension becomes unnecessarily large. Taken together, these results suggest that, in the present same-$K$ setting, representation requirements are not explained by hidden-state count alone, but also depend on the family-specific predictive structure that remains observable after discretization.

## 1 Introduction

Discretization is a common preprocessing step in time-series analysis. Continuous observations are often converted into symbolic sequences before applying sequence models, either for statistical modeling, interpretability, compression, or compatibility with discrete architectures. Examples include price movements transformed into up/down categories, sensor measurements mapped to finite alphabets, and continuous latent signals summarized by symbolic events. Despite this widespread use, the effect of discretization on forecasting is still not well understood. A discretization may preserve the predictive structure of the original process, but it may also destroy it by merging observational histories that imply different future distributions.

This issue is especially important when discretized sequences are modeled by modern sequence learners. In practice, model design is often discussed in terms of nominal architectural size, such as hidden dimension or embedding dimension. However, the difficulty of prediction is not determined solely by the number of latent states or by the superficial size of the observation alphabet. What matters more fundamentally is the predictive structure that remains observable after discretization: how many distinct predictive states survive, how strongly long-range context affects the Bayes-optimal forecast, and how difficult it is for a learner to recover that context sensitivity from finite data. This suggests that discretization should be studied not only as a representation choice, but also as an operation that changes predictive complexity.

In this paper, we study discretization from the viewpoint of predictive states and Hankel-rank-based complexity. Our starting point is the distinction between discretizations that are predictively sufficient and those that induce predictive-state collapse. If a discretized process preserves all information needed for future prediction, then it should retain the essential predictive structure of the original process. If it merges histories with different future laws, then predictive complexity may collapse, and the learner is forced to solve a different and potentially easier problem. This perspective connects discretization to finite-rank pre-

dictive representations and provides a natural language for discussing when coarsening preserves or destroys forecasting-relevant structure.

The main empirical question we address is whether nominal hidden-state cardinality is an adequate proxy for forecasting difficulty after discretization in a controlled same-$K$ setting. To study this in a controlled way, we construct synthetic same-$K$ hidden Markov model families that share the same hidden-state count but differ in Bayes-level context sensitivity. These families are designed so that latent-state cardinality is fixed, while the observable effect of additional context varies across families. This allows us to separate two notions that are often conflated: the size of the latent mechanism and the effective predictive difficulty visible to the learner.

Our experiments show that hidden-state count alone does not determine forecasting difficulty, even within same-$K$ families. One-step prediction accuracy often compresses the differences between families, whereas context-sensitive quantities reveal substantial variation. In particular, the learner-side recovery of Bayes-level context dependence is strongly family-dependent, and its relation to hidden dimension is not monotone. Some families improve as representation size increases up to a moderate range, while others degrade once the model becomes unnecessarily large. These results indicate that, within the present same-$K$ benchmark, representation requirements are not explained by hidden-state count alone, but also depend on the predictive structure that survives discretization and must be recovered from context.

The contribution of this paper is therefore twofold. First, we organize the theoretical viewpoint that links discretization, predictive sufficiency, predictive-state collapse, and finite-rank predictive representation. This clarifies why coarsening can reduce predictive complexity and why nominal state count is, by itself, an incomplete descriptor of forecasting difficulty. Second, we provide controlled same-$K$ experiments showing that, even when latent-state cardinality is fixed, observable predictive difficulty and learner-side representation requirements can differ substantially across families.

More specifically, our contributions are as follows:

1. we formulate the problem of discretization in terms of predictive structure, emphasizing predictive-sufficient discretization and predictive-state collapse under coarsening;

2. we introduce same-$K$ synthetic hidden Markov model families that isolate Bayes-level context effects while holding hidden-state cardinality fixed;

3. we show empirically that hidden-state count alone is insufficient to explain forecasting difficulty, and that context-sensitive recovery behavior can be family-dependent and non-monotone in hidden dimension.

Taken together, these results support a simple message: in controlled discretized same-$K$ settings, the relevant notion of observable forecasting difficulty is not exhausted by hidden-state count alone. A more informative view must also account for the predictive structure preserved by discretization and for the extent to which a learner can recover that structure from context.

The rest of the paper is organized as follows. Section 2 introduces the predictive-state viewpoint and the basic complexity notions used in the paper. Section 3 discusses predictive sufficiency, coarsening, and finite-rank predictive representation. Section 4 presents the experimental setup for the final same-$K$ study. Section 5 reports the main empirical results. Section 6 discusses limitations and scope, and Section 7 concludes.

**Related work.** Our perspective is closely related to several lines of prior work. First, the predictive-state viewpoint is closely connected to computational mechanics, where causal states are defined as equivalence classes of pasts that induce the same future law, and are shown to provide minimal sufficient predictive representations (Shalizi and Crutchfield, 2001). Second, our discussion is related to predictive state representations (PSRs), which replace latent-state descriptions by predictions of future observable events (Littman et al., 2001), and to subsequent work on learning predictive representations from data (Boots et al., 2011). Third, our use of Hankel-rank-based predictive complexity is connected to spectral and linear-algebraic approaches to sequential models, including spectral learning for hidden Markov models (Hsu et al., 2012) and

weighted automata (Balle et al., 2014), as well as unified treatments linking multiplicity automata, observable operator models, and PSRs (Thon and Jaeger, 2015). These works provide an important structural background for viewing finite-rank predictive representations through Hankel-type objects.

Our focus differs from these literatures in two ways. On the theory side, we emphasize the role of discretization itself, asking when coarsening preserves predictive structure and when it induces predictive-state collapse. On the empirical side, rather than comparing model classes on a broad benchmark, we use controlled same-$K$ hidden Markov families to isolate the difference between latent-state cardinality and observable predictive difficulty after discretization. This allows us to ask a different question from standard PSR or spectral-learning work: not only how to identify a predictive representation, but also how discretization changes the effective forecasting problem seen by a finite learner.

Our work is also related to the broader literature on symbolic representations of time series. Discretization and symbolic encoding are widely used for complexity analysis and forecasting, including order-based symbolic methods such as permutation entropy (Bandt and Pompe, 2002). However, much of this literature evaluates symbolic encodings through descriptive complexity or downstream utility, whereas our emphasis is specifically on predictive sufficiency, predictive-state collapse, and representation requirements for sequence learners. Finally, hidden Markov models remain the standard reference framework for latent-state sequence modeling (Rabiner, 1989), and we use them here not as an end in themselves, but as a controlled family generator for studying how observable predictive structure can vary even when hidden-state count is fixed.

## 2 Background and Problem Setup

### 2.1 Discretized time series and predictive states

Let $\{X_t\}_{t\in\mathbb{Z}}$ be a stationary time series taking values in an observation space $\mathcal{X}$. We consider a discretized process $\{S_t\}_{t\in\mathbb{Z}}$ defined by

$$S_t = \phi(X_t),$$

where $\phi : \mathcal{X} \to \mathcal{A}$ is a measurable map into a finite alphabet $\mathcal{A}$. Our interest is not only in the symbolic representation itself, but in how this discretization changes the forecasting problem.

For a past realization $s_{-\infty:t} = (\dots, s_{t-1}, s_t)$, the corresponding predictive law is the conditional distribution of the future,

$$\mathbb{P}(S_{t+1:\infty} \in \cdot \mid S_{-\infty:t} = s_{-\infty:t}).$$

Two pasts are said to be predictively equivalent if they induce the same conditional distribution of the future; this viewpoint is closely related to the causal-state formulation of computational mechanics (Shalizi and Crutchfield, 2001).

**Definition 1** (Predictive equivalence)**.** *Two pasts $p^-$ and $q^-$ are predictively equivalent, written $p^- \sim q^-$, if*

$$\mathbb{P}(S_{t+1:\infty} \in B \mid S_{-\infty:t} = p^-) = \mathbb{P}(S_{t+1:\infty} \in B \mid S_{-\infty:t} = q^-)$$

*for every measurable event $B$ in the future sigma-field.*

The equivalence classes induced by $\sim$ are the predictive states. They summarize exactly the information in the past that is relevant for future prediction. This viewpoint is useful because it separates raw observation complexity from the smaller amount of structure that is actually needed for forecasting.

### 2.2 Predictive complexity and context sensitivity

A central quantity in this paper is the extent to which additional context improves optimal prediction. For a context length $\ell \geq 1$, define the Bayes-optimal one-step log-loss using only the most recent $\ell$ symbols by

$$R_\ell^* = \mathbb{E}[-\log \mathbb{P}(S_{t+1} \mid S_{t-\ell+1:t})].$$

The full-past Bayes risk is

$$R_\infty^* = \mathbb{E}[-\log \mathbb{P}(S_{t+1} \mid S_{-\infty:t})].$$

We then define the context gap at length $\ell$ by

$$\Delta_\ell^* := R_\ell^* - R_\infty^*.$$

A large value of $\Delta_\ell^*$ means that truncating context to length $\ell$ removes information that is genuinely useful for Bayes-optimal prediction. Thus, $\Delta_\ell^*$ serves as a natural measure of delayed predictive relevance.

This perspective is complementary to nominal hidden-state size. Even when two processes are generated by hidden models with the same number of latent states, they may exhibit very different context-gap profiles after discretization. In such cases, the learner faces different observable forecasting tasks despite identical latent-state cardinality.

## 2.3 Hankel-rank-based predictive complexity

To connect predictive structure with finite-dimensional representation, we use the Hankel matrix of the discretized process. Let $\mathcal{A}^*$ denote the set of finite words over $\mathcal{A}$. For $u, v \in \mathcal{A}^*$, define

$$H(u, v) = \mathbb{P}\big(S_{1:|u|} = u,\ S_{|u|+1:|u|+|v|} = v\big).$$

The rank of this bi-infinite matrix, when finite, is a standard structural measure in spectral and linear-representation approaches to sequential models (Hsu et al., 2012; Balle et al., 2014; Thon and Jaeger, 2015).

We do not claim in this paper that Hankel rank alone completely determines practical learning behavior. Rather, we use it as a structural viewpoint: discretization can preserve predictive distinctions, or collapse them, and such collapse is naturally reflected in reduced predictive complexity.

## 2.4 Problem statement

The main question of this paper is the following: when continuous or latent dynamics are converted into a symbolic process, what determines the representation size required by a learner to recover forecasting-relevant context? Our working hypothesis is that hidden-state count alone is insufficient. Instead, what matters is the predictive structure that survives discretization, as reflected by Bayes-level context sensitivity and related finite-dimensional predictive structure.

# 3 Predictive Sufficiency, Coarsening, and Representation Requirements

## 3.1 Predictive-sufficient discretization

A discretization should be considered successful for forecasting only if it preserves the information in the original process that is relevant for predicting the future.

**Definition 2** (Predictive-sufficient discretization)**.** *Let $S_t = \phi(X_t)$. We say that the discretization is predictively sufficient if, for every $t$,*

$$\mathbb{P}(X_{t+1:\infty} \in B \mid X_{-\infty:t}) = \mathbb{P}(X_{t+1:\infty} \in B \mid S_{-\infty:t})$$

*for every measurable future event $B$.*

If this condition holds, then the symbolic past retains all information needed for future prediction in the original process. If it fails, then discretization has merged pasts that are distinct from the viewpoint of forecasting.

## 3.2 Predictive-state collapse under coarsening

Coarsening can reduce predictive complexity by collapsing predictive states. At an intuitive level, this happens when two distinct observational histories become indistinguishable after discretization, even though they imply different future laws before coarsening.

**Theorem 1** (Effect of discretization on predictive complexity)**.** *Let $\{X_t\}$ be a stationary process and let $\{S_t\}$ be defined by $S_t = \phi(X_t)$. Let $r_X$ and $r_S$ denote the Hankel ranks of $\{X_t\}$ and $\{S_t\}$, respectively, whenever these ranks are well defined and finite. Then:*

1. *$r_S \leq r_X$;*

2. *if the discretization is predictively sufficient, then $r_S = r_X$.*

The first statement formalizes the fact that coarsening cannot create new predictive distinctions. The second states that if discretization preserves all future-relevant information, then predictive complexity is preserved rather than collapsed. A proof sketch is given in Appendix A, under the finite-rank representation assumptions adopted in this paper.

This theorem should be interpreted as a structural statement, not as a complete learning theorem. It explains why discretization can change the intrinsic forecasting problem, but it does not by itself determine how easily a finite learner will recover the remaining structure from finite samples.

### 3.3 Why hidden-state count is not enough

A hidden Markov model with $K$ latent states does not necessarily induce a unique observable level of predictive difficulty. Even with fixed $K$, the organization of transition structure and emissions can produce different observable context dependencies after discretization. This is the key reason we focus on same-$K$ families.

**Proposition 1** (Same-$K$ does not imply same observable difficulty)**.** *There exist families of hidden Markov models with the same hidden-state cardinality $K$ such that the resulting discretized processes have different Bayes context-gap profiles $\{\Delta_\ell^*\}_{\ell \geq 1}$. Consequently, fixed latent-state count does not determine observable forecasting difficulty.*

This proposition is conceptually simple but important for experimental design. If two same-$K$ families differ substantially in their context-gap profiles, then a learner with a fixed architecture may behave very differently across them, even though the nominal hidden-state size is identical.

### 3.4 Representation requirements and learner-side recovery

The theory above motivates a distinction between two questions. The first is a Bayes-level question: how much future-relevant information is lost when context is truncated? The second is a learner-side question: how much of this Bayes-level context dependence is actually recovered by a trained model of a given size?

This distinction is essential in our experiments. A process may exhibit a large Bayes context gap, yet a given learner may fail to recover it. Conversely, a family with a small Bayes gap may saturate quickly, so that increasing hidden dimension adds little benefit and may even hurt due to estimation or optimization effects. This is the sense in which representation requirements can be family-dependent and non-monotone in model dimension.

## 4 Experimental Setup

This section describes the construction of the final same-$K$ benchmark, the learner-side training protocol, and the evaluation metrics used in the experiments.

### 4.1 Final same-$K$ family selection

Our goal is to compare processes that share the same latent-state cardinality but differ in observable Bayes-level context sensitivity after discretization. To this end, we first generated candidate hidden Markov model families with a fixed hidden-state count $K$ (Rabiner, 1989), and then evaluated their Bayes-level context-response curves.

From these candidates, we selected two representative groups for the final study: a *middle-complexity family* and a *highest-complexity family*. The two groups were chosen so that they share the same hidden-state cardinality while exhibiting clearly different delayed context effects at the Bayes level. In particular, the highest-complexity family was selected to have a substantially larger delayed Bayes gap than the middle-complexity family.

Each group contains multiple independently generated candidate processes. The purpose of this design is to reduce dependence on a single hand-picked process and to evaluate learner-side behavior across a small but nontrivial family of same-$K$ instances. For the main-text comparison, we focus on the middle- and highest-complexity groups because they provide the clearest same-$K$ contrast in Bayes-level context sensitivity. For completeness, supplementary three-family results including the low-complexity group are reported in Appendix E.

## 4.2 Symbolic prediction task

For each selected process, we discretized the observable sequence into a finite symbol sequence and considered one-step prediction of the next symbol from a truncated context. Bayes-level reference losses were computed for context lengths

$$h \in \{0, 1, 2, 3, 4\},$$

yielding context-response curves $L^{\text{Bayes}}(h)$. Here $h = 0$ denotes the no-context baseline, in which the predictor uses no past symbols and relies only on the marginal next-symbol law. These curves serve as the target predictive structure that the learner is expected to recover.

On the learner side, we evaluated one-step prediction using context lengths $h \in \{1, 2, 3, 4\}$, and recorded the corresponding test cross-entropies and accuracies. This allows direct comparison between Bayes-level context sensitivity and learner-side recovery behavior.

## 4.3 Model and optimization

Our primary learner is a gated recurrent unit (GRU) predictor (Cho et al., 2014) trained for next-symbol prediction. The main hidden-dimension sweep uses

$$d \in \{32, 64, 128, 256\}.$$

For the final hidden-dimension comparison, the training-set size is fixed at $n_{\text{train}} = 256$, and the default training budget is 12 epochs.

To reduce variance due to optimization randomness, we train multiple independent seeds for each setting and aggregate the results over seeds. The main hidden-dimension sweep is therefore interpreted through mean behavior and seed-level variability rather than through a single best run.

Unless otherwise noted, the main hidden-dimension comparison uses a fixed training protocol across families and hidden dimensions. For each selected process, we generate symbolic sequences of length 300, with a maximum training pool of 512 sequences, together with 32 validation sequences and 64 test sequences. In the main hidden-dimension sweep, the model is trained on the first $n_{\text{train}} = 256$ training sequences for 12 epochs using Adam with learning rate $10^{-3}$ and batch size 16, and results are averaged over 7 independent training seeds. The GRU predictor uses an embedding layer of dimension $d$, a single GRU layer with hidden size $d$, and a linear output layer over the observation vocabulary. Training is truncation-aware: for each minibatch, the effective context length is sampled from $\{1, 2, 3, 4\}$, while evaluation is carried out separately for each fixed context length $h \in \{1, 2, 3, 4\}$. Although validation sequences are generated as part of the fixed dataset construction, the main sweep does not use validation-based checkpoint selection or early stopping; models are evaluated after the prescribed epoch budget. Further reproducibility details for the final same-$K$ study are given in Appendix C.

### 4.4 Main evaluation metrics

A central point of this paper is that one-step accuracy alone is often too coarse to reveal differences between same-$K$ families. We therefore evaluate learner-side behavior using context-sensitive summaries.

**Bayes context gap.** For a given family, the delayed Bayes gain from additional context is summarized by

$$\Delta_{1\to4}^{\mathrm{Bayes}} = L^{\mathrm{Bayes}}(1) - L^{\mathrm{Bayes}}(4),$$

and similarly for

$$\Delta_{2\to4}^{\mathrm{Bayes}} = L^{\mathrm{Bayes}}(2) - L^{\mathrm{Bayes}}(4).$$

Larger values indicate that longer context provides greater Bayes-optimal improvement.

**Empirical learner-side context gap.** For a trained model, we define

$$\Delta_{1\to4}^{\mathrm{emp}} = L^{\mathrm{model}}(1) - L^{\mathrm{model}}(4),$$

with an analogous definition for $2 \to 4$. This measures how much the learned predictor benefits from additional context.

**Recovery ratio.** To quantify how much of the Bayes-level gain is recovered by the learner, we use

$$RR_{1\to4} = \frac{\Delta_{1\to4}^{\mathrm{emp}}}{\Delta_{1\to4}^{\mathrm{Bayes}}}.$$

A value near 1 indicates that the learner recovers most of the delayed context effect visible at the Bayes level, whereas smaller or negative values indicate poor recovery.

**Shape RMSE.** To compare the full context-response shape rather than a single gap alone, we compute a centered shape RMSE. Specifically, both the learned and Bayes context-response curves are re-expressed relative to context length 4, yielding

$$\big(L(1) - L(4),\ L(2) - L(4),\ L(3) - L(4),\ 0\big).$$

Shape RMSE is then defined as the root mean squared deviation between the resulting learned and Bayes vectors. Lower values indicate better recovery of family-specific context dependence.

**One-step accuracy.** We also report one-step test accuracy for each context length. However, this is treated as a secondary metric, since our main interest is not only marginal predictive quality but also the recovery of delayed context sensitivity.

Formal definitions of these metrics are given in Appendix B.

### 4.5 Auxiliary sweeps

In addition to the main hidden-dimension sweep, we performed three auxiliary studies to assess the robustness of the observed behavior.

First, we varied the number of training epochs over

$$\{4, 8, 12, 24, 48\}$$

to examine whether non-monotone recovery is merely an early-training artifact.

Second, we varied the training-set size over

$$\{32, 64, 128, 256, 512\}$$

to evaluate the effect of sample size on learner-side recovery.

Third, we examined several regularization settings, including weight decay, dropout, and their combination, in order to test whether the degradation at large hidden dimension can be mitigated by more stable optimization.

These auxiliary sweeps are not the main result of the paper. Their role is to clarify the interpretation of the hidden-dimension effect and to distinguish family-specific structure from purely optimization-driven variation.

### 4.6 Questions tested

The experiments are designed to answer the following questions:

1. Do same-$K$ families exhibit clearly different Bayes-level context-response curves?

2. Does one-step predictive accuracy adequately reveal these differences?

3. How does learner-side recovery vary with hidden dimension?

4. Is the relation between hidden dimension and recovery monotone across families?

5. To what extent are these trends affected by epoch budget, sample size, and regularization?

## 5 Main Experimental Results

In this section, we report the main empirical results for the final same-$K$ study. For the final comparison, we retained two representative families, denoted the middle-complexity family and the highest-complexity family. These families share the same hidden-state cardinality, but differ in Bayes-level context sensitivity after discretization. The main text emphasizes the middle- versus highest-complexity contrast, while Appendix E reports supplementary three-family results including the low-complexity group.

### 5.1 Bayes-level separation among same-$K$ families

We first examine whether the selected same-$K$ families are genuinely different at the Bayes level. Figure 1 shows the Bayes cross-entropy curves as a function of context length.

The two families are clearly separated. For the highest-complexity family, the Bayes cross-entropy decreases from 2.0332 at context length 1 to 2.0265 at context length 4, yielding a delayed Bayes gap of approximately 0.0067. By contrast, for the middle-complexity family, the corresponding decrease is from 2.0337 to 2.0308, with a delayed Bayes gap of about 0.0029. The same pattern is visible for the context-2 to context-4 gap: approximately 0.00335 for the highest-complexity family versus 0.00112 for the middle-complexity family.

These results confirm that fixed hidden-state count does not determine Bayes-level predictive difficulty. Even within same-$K$ families, the observable usefulness of additional context can differ substantially.

For completeness, the supplementary three-family comparison in Appendix E shows an ordered Bayes-level separation across low-, middle-, and high-complexity same-$K$ families. Averaged over retained candidates, the mean delayed Bayes gaps $\Delta_{1\to4}^{\mathrm{Bayes}}$ are approximately 0.00232, 0.00292, and 0.00673, respectively. Thus, the additional low-complexity family does not weaken the main conclusion; rather, it reinforces the point that Bayes-level context sensitivity can vary substantially even when hidden-state cardinality is fixed.

### 5.2 Learner-side recovery depends strongly on hidden dimension

We next study how well GRU predictors recover this Bayes-level context dependence. Figure 2 plots the recovery ratio as a function of hidden dimension.

The dependence on hidden dimension is strongly family-specific. For the highest-complexity family, the mean recovery ratio improves from $0.687 \pm 0.220$ at hidden dimension 32 to a peak of $0.781 \pm 0.184$ at hidden

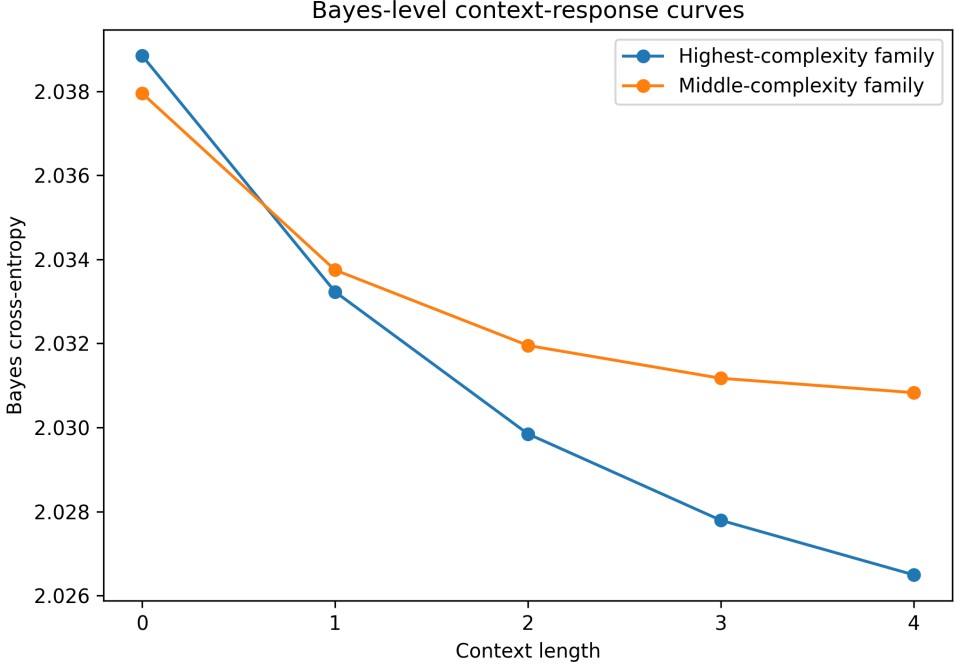

Figure 1: Estimated Bayes-level context-response curves for the selected same-$K$ families. Even with fixed hidden-state count, the gain from additional context differs across families.

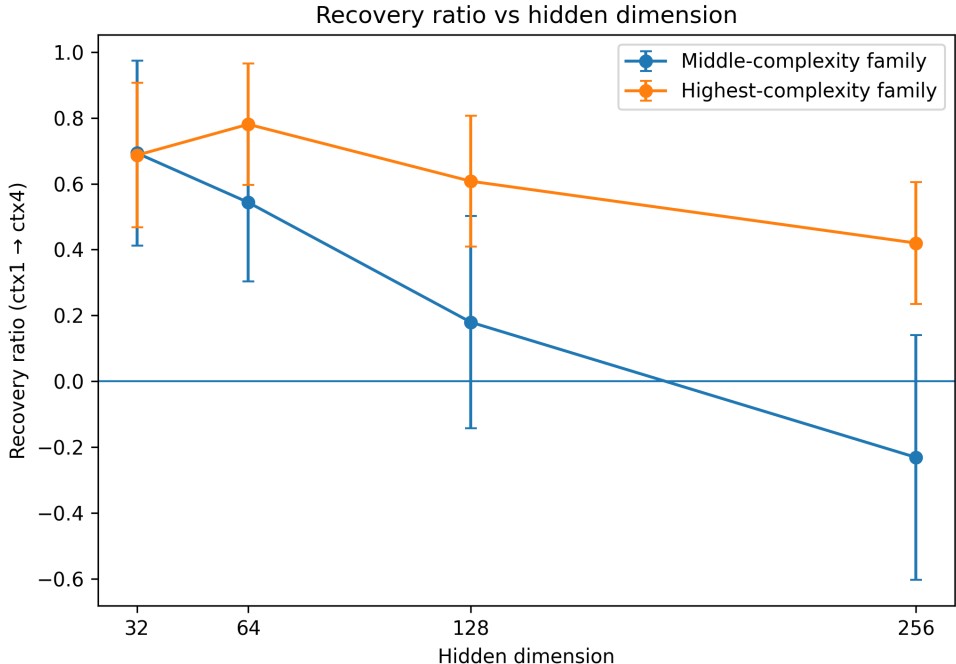

Figure 2: Recovery ratio as a function of hidden dimension for the selected same-$K$ families. Error bars show standard deviation over training seeds.

dimension 64, and then declines to $0.608 \pm 0.198$ at 128 and $0.420 \pm 0.185$ at 256. Thus, larger hidden dimension helps up to an intermediate scale, but the effect is clearly non-monotone.

For the middle-complexity family, the non-monotonicity is even more pronounced. The mean recovery ratio is $0.693 \pm 0.281$ at hidden dimension 32, drops to $0.543 \pm 0.239$ at 64, falls further to $0.179 \pm 0.322$ at 128, and becomes negative, $-0.231 \pm 0.372$, at 256. Hence, for this family, increasing hidden dimension beyond a moderate range substantially harms learner-side recovery of Bayes-level context sensitivity.

Taken together, these results show that hidden-state count alone is not a reliable guide to representation size. Even with the same latent-state cardinality, different families can require different hidden dimensions for effective context recovery.

### 5.3 Curve-shape fidelity also deteriorates outside the appropriate range

To evaluate not only the magnitude of recovery but also the fidelity of the entire context-response curve, we measure the shape RMSE between the learned curve and the Bayes curve. Figure 3 summarizes this comparison.

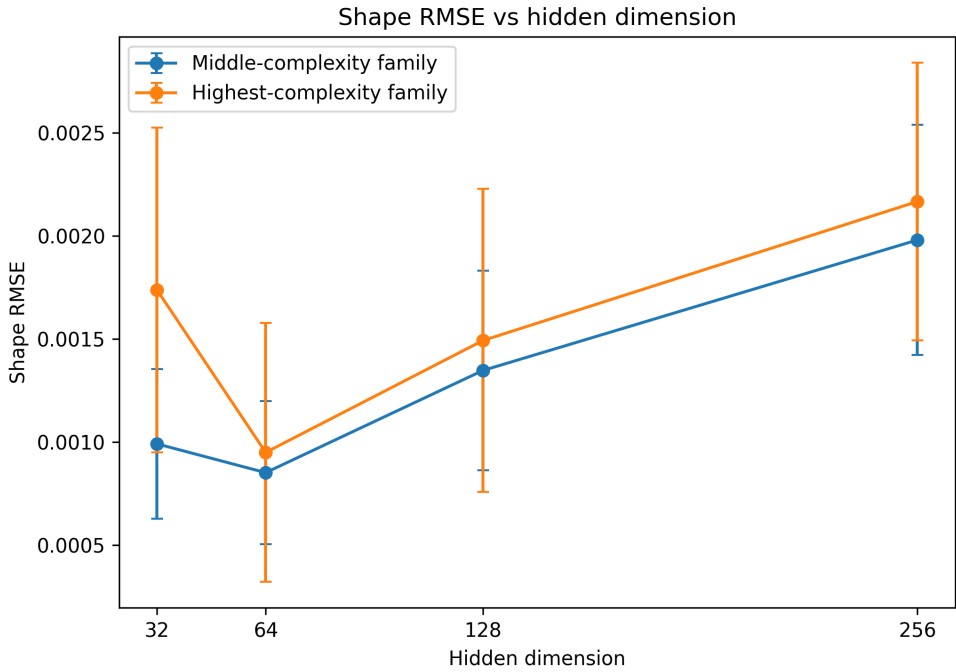

Figure 3: Shape RMSE between learned and Bayes context-response curves as a function of hidden dimension. Lower values indicate better recovery of family-specific context dependence.

The shape-based analysis supports the same conclusion. For the highest-complexity family, the lowest mean shape RMSE is attained at hidden dimension 64, where it reaches approximately $0.00095 \pm 0.00063$. Performance deteriorates at larger dimensions, with mean shape RMSE increasing to about $0.00149$ at 128 and $0.00217$ at 256.

For the middle-complexity family, the best shape fidelity is also obtained in the small-to-moderate range. The mean shape RMSE is approximately $0.00099 \pm 0.00036$ at hidden dimension 32 and $0.00085 \pm 0.00035$ at 64, but worsens to $0.00135 \pm 0.00048$ at 128 and $0.00198 \pm 0.00056$ at 256. Thus, the learned context-response shape becomes less faithful once the hidden dimension moves beyond the range appropriate for the family.

The supplementary three-family analysis in Appendix E also supports the shape-based interpretation. For the middle- and high-complexity families, the best shape fidelity is attained at hidden dimension 64. For the low-complexity family, the minimum shape RMSE is attained at 128, although the difference between 64 and

128 is comparatively modest. Overall, the three-family comparison still supports the main conclusion that the relation between hidden dimension and learner-side recovery is family-dependent rather than uniformly monotone.

### 5.4 Interpretation

The main empirical message of this section is threefold.

First, same-$K$ families can differ substantially at the Bayes level. The selected highest-complexity family exhibits much stronger delayed context effects than the middle-complexity family, despite identical hidden-state cardinality.

Second, one-step predictive performance alone does not fully capture this difference. Although the highest-complexity family tends to achieve somewhat higher context-4 test accuracy than the middle-complexity family, the absolute differences in one-step accuracy are modest compared with the much clearer separation seen in Bayes gaps, recovery ratios, and shape RMSE.

Third, learner-side recovery is family-dependent and non-monotone in hidden dimension. For the highest-complexity family, the best recovery is obtained around hidden dimension 64. For the middle-complexity family, the best range is smaller, and large hidden dimensions substantially degrade recovery. Accordingly, representation requirements cannot be inferred from hidden-state count alone. What matters is the predictive structure that remains observable after discretization and the extent to which a finite learner can recover it from context.

Overall, these results support the central claim of the paper: nominal hidden-state cardinality is an incomplete descriptor of forecasting difficulty, and context-sensitive predictive complexity provides a more informative perspective.

The supplementary three-family results in Appendix E are consistent with this interpretation: they preserve the same main picture while showing that the low-complexity family occupies a weaker-context regime rather than invalidating the family-dependent pattern.

## 6 Limitations and Scope

This paper has several limitations.

First, the experiments are conducted on synthetic hidden Markov families. This is deliberate, since the goal is controlled comparison, but it means that the present conclusions should be interpreted primarily as structural evidence rather than as a claim about all real-world time series.

Second, the learner-side analysis is centered on GRU models. This choice is useful for isolating representation-size effects under a stable training pipeline, but it does not by itself establish that the same quantitative behavior must hold for all sequence-model architectures.

Third, while the theoretical discussion is motivated by predictive sufficiency and Hankel-rank-based complexity, the experiments do not directly estimate full infinite Hankel rank. Instead, they probe observable predictive difficulty through Bayes-level context gaps and learned recovery behavior. Thus, the connection between structural predictive complexity and practical model dimension should be viewed here as strongly motivated rather than fully closed.

Finally, the notion of an "effective predictive dimension" is used in this paper as an interpretive concept. It is intended to summarize family-dependent representation requirements, but it is not yet presented as a unique or fully formal invariant. A more rigorous characterization remains an important direction for future work.

# 7 Conclusion

We studied discretization as an operation that changes predictive structure, rather than as a purely representational preprocessing step. From the viewpoint of predictive states, coarsening may preserve future-relevant information or collapse predictive distinctions, and this naturally affects predictive complexity.

Motivated by this perspective, we introduced same-$K$ hidden Markov model families that fix latent-state cardinality while varying Bayes-level context sensitivity after discretization. The resulting experiments, including the supplementary three-family comparison, show that hidden-state count alone does not determine observable forecasting difficulty within the present benchmark. They also show that learner-side recovery of Bayes-level context dependence is family-dependent and non-monotone in hidden dimension.

The main implication is that, in our controlled same-$K$ experiments, representation requirements for discretized time series cannot be inferred from hidden-state count alone. A more informative account must also consider the predictive structure preserved under discretization and the extent to which a finite learner can recover it from context.

Future work includes extending the analysis to other sequence-model classes, strengthening the connection to finite-rank predictive representations, and testing whether the same qualitative phenomena persist beyond the present synthetic benchmark.

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

# A Proof Sketch for Theorem 1

In this appendix, we give a proof sketch of Theorem 1. The purpose is not to establish a full general theory of discretization for arbitrary stochastic processes, but to justify the rank-monotonicity statement used in

the main text under the finite-rank representation assumptions adopted in this paper. In particular, the argument is intended for settings in which the relevant Hankel ranks are well defined and finite, and in which finite-rank predictive representations are taken as the structural object of interest.

Let $\{X_t\}$ be a stationary process over an observation space $\mathcal{X}$, and let $\{S_t\}$ be the discretized process defined by

$$S_t = \phi(X_t),$$

where $\phi : \mathcal{X} \to \mathcal{A}$ maps observations into a finite alphabet $\mathcal{A}$. Write $H_X$ and $H_S$ for the Hankel matrices of the original and discretized processes, respectively.

**Step 1: the discretized Hankel matrix is an aggregation of the original one.** For any finite words $u, v \in \mathcal{A}^*$, define

$$H_S(u,v) = \mathbb{P}\big(S_{1:|u|} = u,\ S_{|u|+1:|u|+|v|} = v\big).$$

Each symbolic word $u$ corresponds to a collection of observation-level words whose image under $\phi$ is equal to $u$. Accordingly, $H_S(u,v)$ can be written as the sum of the corresponding entries of $H_X$ over all observation-level words that coarsen to $u$ and $v$. Thus, there exist linear aggregation operators $L$ and $R$ such that

$$H_S = L H_X R.$$

Since left and right multiplication by linear maps cannot increase matrix rank, we obtain

$$\mathrm{rank}(H_S) \leq \mathrm{rank}(H_X).$$

This proves the first claim.

**Step 2: predictive sufficiency prevents rank collapse under the finite-rank representation assumptions.** Assume now that the discretization is predictively sufficient, that is,

$$\mathbb{P}(X_{t+1:\infty} \in B \mid X_{-\infty:t}) = \mathbb{P}(X_{t+1:\infty} \in B \mid S_{-\infty:t})$$

for every measurable future event $B$. Then the symbolic past retains all future-relevant information contained in the original past. In particular, predictive distinctions present in the original process are not lost after discretization.

Under the finite-rank assumptions adopted here, the Hankel rank coincides with the dimension of a minimal linear predictive representation. Predictive sufficiency therefore implies that the predictive representation induced by the discretized process is not a collapsed version of that of the original process: the two processes encode the same predictive distinctions, up to relabeling through the sufficient symbolic history. Hence the minimal predictive dimension is preserved, and therefore

$$\mathrm{rank}(H_S) = \mathrm{rank}(H_X).$$

This proves the second claim at the level of generality intended in the main text.

**Remark.** The argument above is intentionally stated at the level needed in the present paper. A more complete treatment would require a fuller discussion of measurability, minimality of predictive representations, and the precise class of processes for which the finite-rank equivalence is valid.

## B Metric Definitions Used in Section 4

This appendix records the evaluation quantities used in the experiments in a notation that is independent of implementation details.

For a family $f$ and context length $h$, let

$$L_f^{\mathrm{Bayes}}(h)$$

denote the estimated Bayes one-step cross-entropy at context length $h$. Similarly, let

$$L_{f,d}^{\text{model}}(h)$$

denote the corresponding test cross-entropy for a trained model with hidden dimension $d$.

In the main text, when the family and hidden dimension are clear from context, we use the abbreviated notation

$$L^{\text{Bayes}}(h), \qquad L^{\text{model}}(h).$$

**Bayes context gap.**   The delayed Bayes gain from longer context is summarized by

$$\Delta_{f,1\to4}^{\text{Bayes}} = L_f^{\text{Bayes}}(1) - L_f^{\text{Bayes}}(4),$$

and likewise

$$\Delta_{f,2\to4}^{\text{Bayes}} = L_f^{\text{Bayes}}(2) - L_f^{\text{Bayes}}(4).$$

**Empirical learner-side context gap.**   For a trained model, define

$$\Delta_{f,d,1\to4}^{\text{emp}} = L_{f,d}^{\text{model}}(1) - L_{f,d}^{\text{model}}(4),$$

and analogously for $2 \to 4$.

**Recovery ratio.**   The recovery ratio used in the main text is

$$RR_{f,d,1\to4} = \frac{\Delta_{f,d,1\to4}^{\text{emp}}}{\Delta_{f,1\to4}^{\text{Bayes}}}.$$

A value near 1 indicates that the learner recovers most of the delayed context effect visible at the Bayes level.

**Shape RMSE.**   To compare full context-response curves, we use a centered representation relative to context length 4. Define

$$\widetilde{L}_f^{\text{Bayes}} = \left(L_f^{\text{Bayes}}(1) - L_f^{\text{Bayes}}(4), L_f^{\text{Bayes}}(2) - L_f^{\text{Bayes}}(4), L_f^{\text{Bayes}}(3) - L_f^{\text{Bayes}}(4), 0\right),$$

and similarly

$$\widetilde{L}_{f,d}^{\text{model}} = \left(L_{f,d}^{\text{model}}(1) - L_{f,d}^{\text{model}}(4), L_{f,d}^{\text{model}}(2) - L_{f,d}^{\text{model}}(4), L_{f,d}^{\text{model}}(3) - L_{f,d}^{\text{model}}(4), 0\right).$$

We then define

$$\text{ShapeRMSE}_{f,d} = \left(\frac{1}{4}\sum_{j=1}^{4}\left(\widetilde{L}_{f,d,j}^{\text{model}} - \widetilde{L}_{f,j}^{\text{Bayes}}\right)^2\right)^{1/2}.$$

Lower values indicate better recovery of the family-specific context-response shape.

## C   Reproducibility Details for the Final same-$K$ Study

This appendix summarizes the concrete data-construction and training details used in the final same-$K$ study.

The benchmark is constructed from hidden Markov model families with fixed hidden-state cardinality $K = 4$ and observation vocabulary size 8. During candidate generation, we use three family roles, denoted *low-rank*, *medium-rank*, and *high-rank*. For each family role, 60 candidate seeds are explored, with a prefilter pool size of 16 and 2500 search trials in the calibration stage. The calibration stage retains 4 selected candidate

processes per family role. In the final comparison reported in the main text, we use the retained *middle-complexity* and *highest-complexity* groups, which correspond to the retained *medium-rank* and *high-rank* candidate families, respectively.

For each selected process, we generate symbolic sequences of length 300. The fixed dataset construction produces a maximum training pool of 512 sequences, together with 32 validation sequences and 64 test sequences. In the main hidden-dimension sweep, the learner is trained on the first $n_{\text{train}} = 256$ training sequences for 12 epochs, with hidden dimension

$$d \in \{32, 64, 128, 256\},$$

batch size 16, and the Adam optimizer with learning rate $10^{-3}$. Results are averaged over 7 independent training seeds.

The GRU predictor consists of an embedding layer of dimension $d$, a single-layer GRU with hidden size $d$, and a linear output layer over the observation vocabulary. Training is truncation-aware: in each minibatch, the effective context length is sampled from

$$\{1, 2, 3, 4\},$$

and the model is trained for one-step next-symbol prediction. Evaluation is performed on the fixed test set at each context length

$$h \in \{1, 2, 3, 4\}.$$

Although validation sequences are generated as part of the fixed dataset construction, the main hidden-dimension sweep does not use validation-based checkpoint selection or early stopping; models are evaluated after the prescribed epoch budget. No additional regularization is used in the main hidden-dimension sweep.

The auxiliary sweeps use the same basic training pipeline with modified control variables. The epoch sweep uses 5 training seeds, $n_{\text{train}} = 256$, and reports checkpoints at epochs

$$\{4, 8, 12, 24, 48\}.$$

The training-size sweep uses 5 seeds, training sizes

$$\{32, 64, 128, 256, 512\},$$

and 24 training epochs. The regularization sweep uses hidden dimensions

$$\{64, 128, 256\},$$

5 seeds, $n_{\text{train}} = 256$, and 24 epochs. The regularization grid consists of the following settings:

$$\texttt{none}, \quad \texttt{wd1e-4}, \quad \texttt{wd1e-3}, \quad \texttt{drop0.1}, \quad \texttt{drop0.2}, \quad \texttt{wd1e-4\_drop0.1}.$$

Equivalently, this corresponds to weight decay and dropout configurations

$$(0, 0, 0), \ (10^{-4}, 0, 0), \ (10^{-3}, 0, 0), \ (0, 0.1, 0.1), \ (0, 0.2, 0.2), \ (10^{-4}, 0.1, 0.1),$$

where each triple denotes

$$(\text{weight decay, embedding dropout, output dropout}).$$

## D  Additional Interpretation of the Final same-$K$ Study

The main text emphasizes three empirical facts: (i) same-$K$ families differ at the Bayes level, (ii) one-step accuracy alone under-resolves these differences, and (iii) learner-side recovery is family-dependent and non-monotone in hidden dimension. The purpose of this appendix is to clarify the scope of that interpretation.

First, the results do not claim that Hankel rank itself was directly estimated or recovered. Rather, the experiments probe observable consequences of predictive complexity through Bayes-level context sensitivity and learner-side recovery behavior.

Second, the observed non-monotonicity should be interpreted relative to the present training regime. The auxiliary sweeps suggest that epoch budget, sample size, and regularization influence how sharply degradation appears at large hidden dimension. Thus, the main claim is not that larger models are intrinsically worse, but that hidden-state count alone does not determine the representation size needed for effective recovery of family-specific context dependence.

Third, the same-$K$ design is valuable precisely because it separates nominal latent-state cardinality from observable predictive difficulty after discretization. This is the main empirical lesson of the paper.

## E    Supplementary Three-Family Results

The main text focuses on the middle- versus highest-complexity contrast because it provides the clearest same-$K$ comparison in Bayes-level context sensitivity. For completeness, this appendix reports supplementary results including the low-complexity family.

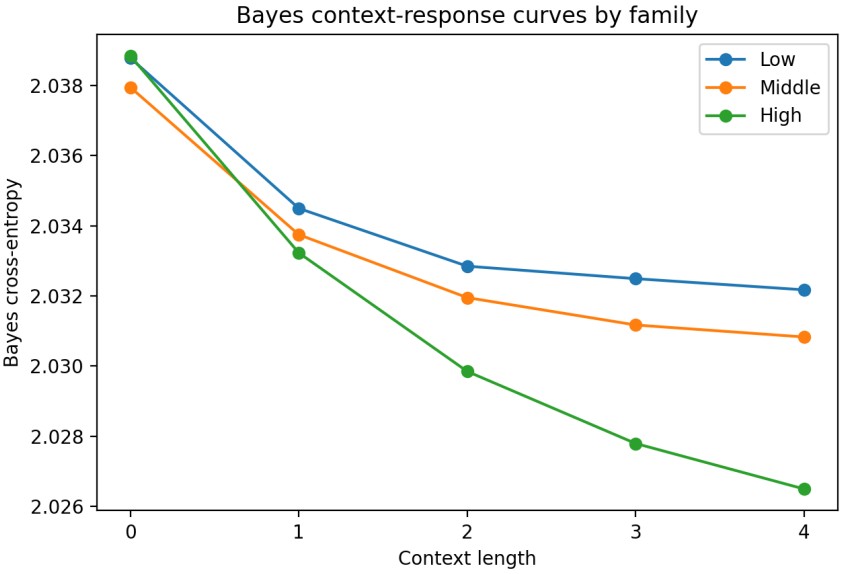

Figure 4: Bayes-level context-response curves for the supplementary three-family same-$K$ comparison. The delayed context effect increases from the low-complexity family to the middle-complexity family and is largest for the high-complexity family, confirming that Bayes-level context sensitivity can vary substantially even when hidden-state cardinality is fixed.

Figure 4 shows the Bayes-level context-response curves for the low-, middle-, and high-complexity families. The three-family comparison exhibits an ordered separation: the high-complexity family has the largest delayed Bayes effect, the middle-complexity family occupies an intermediate regime, and the low-complexity family shows the weakest context dependence. Averaged over retained candidates, the mean delayed Bayes gaps $\Delta_{1 \to 4}^{\mathrm{Bayes}}$ are approximately 0.00232, 0.00292, and 0.00673 for the low-, middle-, and high-complexity families, respectively.

Figure 5 shows the corresponding learner-side recovery ratios as a function of hidden dimension. The high-complexity family again attains its strongest mean recovery around hidden dimension 64, whereas the middle-complexity family is best at a smaller hidden dimension and deteriorates thereafter. The low-complexity family shows a comparatively unstable recovery ratio. This should be interpreted cautiously: because its Bayes-level delayed context gap is small, ratio-based summaries are more sensitive to moderate absolute fluctuations in learner-side performance.

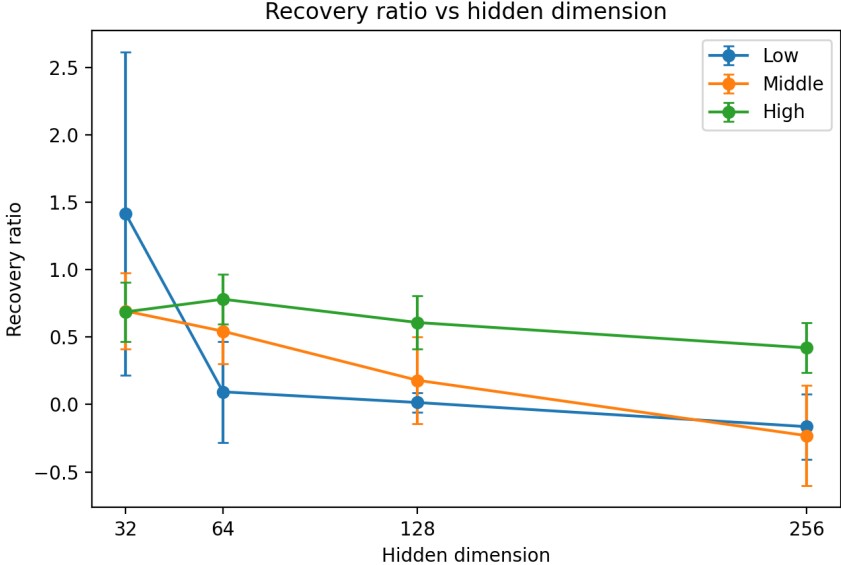

Figure 5: Recovery ratio as a function of hidden dimension for the supplementary three-family comparison. The high-complexity family is best recovered around hidden dimension 64, while the low-complexity family exhibits greater ratio instability because its Bayes-level delayed context gap is small.

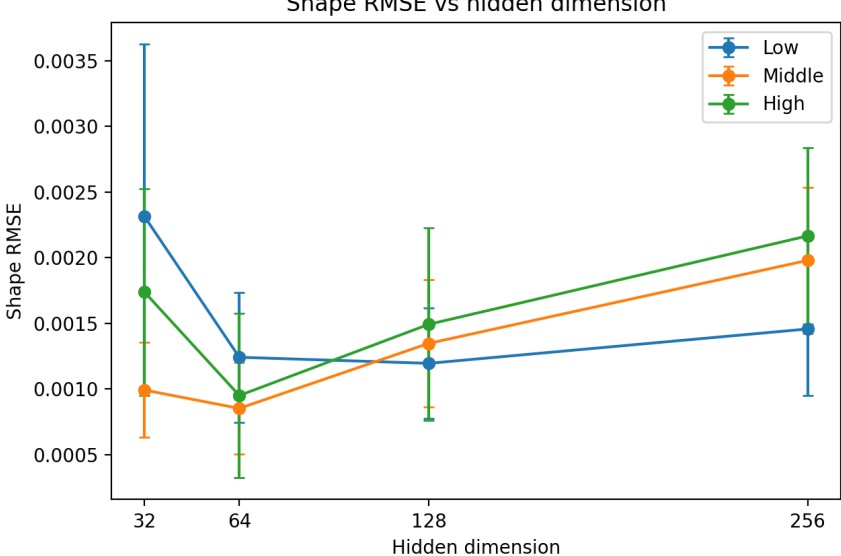

Figure 6: Centered shape RMSE between learned and Bayes context-response curves for the supplementary three-family comparison. Lower values indicate better recovery of family-specific context dependence. The middle- and high-complexity families are best matched around hidden dimension 64, whereas the low-complexity family is flatter and attains a similar minimum at a slightly larger dimension.

Figure 6 reports the centered shape RMSE between the learned and Bayes context-response curves. For the middle- and high-complexity families, the best shape fidelity is attained at hidden dimension 64. For the low-complexity family, the minimum is attained at 128, although the difference between 64 and 128 is comparatively modest. Taken together, these supplementary results reinforce the main interpretation of the paper: same-$K$ families can differ systematically in Bayes-level context sensitivity, and learner-side recovery remains family-dependent and non-monotone in hidden dimension.

