# OpenReview forum: "Discretization and Predictive Complexity of Time Series"
_TMLR — Rejected by TMLR_

### Review · Reviewer_bY6Y · 2026-05-24

**Summary Of Contributions:**

Either I am grossly misunderstanding the paper at a fundamental level, but I cannot see any impactful contributions made by the paper. The authors do not make any novel contributions to the area of time-series predictions. There are a total of nine (9) references with the latest being from 2015 --- surely, relevant work has been done in the area in the last 11 years.

**Audience:**

No

**Audience Explanation:**

Again, the work does not make any impactful contributions. The paper is a basic/trivial investigation into the effects of discretization/truncation of continuous stream of data on next-step predictions.

**Broader Impact Concerns:**

Not applicable.

**Claims And Evidence:**

No

**Claims Explanation:**

No comments.

**Requested Changes:**

No changes.

---

> ### Author Response · Authors · 2026-05-25
> **Acknowledgement and planned response**
>
> We thank the reviewer for reading the paper.
> We will wait for the remaining reviews and then provide a consolidated response and revision.
> In particular, we plan to clarify the intended contribution of the paper and expand the related-work discussion
> to better situate the work relative to recent time-series and sequence-modeling literature.

---

### Review · Reviewer_CTtd · 2026-05-26

**Summary Of Contributions:**

This paper studies state-discretization in time series: i.e., reducing the state $x_t \in \mathbb{R}^d$ in a time series to a discretized state $s_t \in \mathcal{A}$ for some finite alphabet $\mathcal{A}$. The main tool of study here is Hankel rank, i.e., the rank of the infinite matrix describing joint probabilities of words of length $u$ to length $u+v$ words in the alphabet $\mathcal{A}$ appearing in the sequence. The authors additionally study the effect of the "Bayes context gap," the log-loss explained by truncation of the infinite time series. The main message is that the cardinality of $\mathcal{A}$ is insufficient to understand the effect of discretization. These are studied empirically in a "same-K" family of hidden Markov models.

I find the basic problem --- studying the effect of state discretization --- interesting. Unfortunately, I find that there are several deficiencies in the submitted manuscript that prevent me from recommending acceptance. I list a few below:

- **[Rigor]** The main theoretical contribution of the paper is its Theorem 1. This theorem is essentially trivial, to my reading. Despite this, only a proof sketch is provided. Proposition 1, on the other hand, has no proof presented.
- **[Experiments]** The paper is motivated as studying generic discretization in time series, but the presented experiments are for two families of HMMs. This is insufficient to understand the general theory and its implications. There are many addition limitations within this context, i.e., only using very small GRUs in two families.
- **[Writing]** The writing is rather curt, and includes many undefined terms; for example, "same-$K$ family" appears many times before it is actually defined.
- **[References]** There are no references from the last ten years; this is highly unusual for a machine learning paper, and in my opinion insufficient for understanding its place in the literature. There are many ways to contextualize this work, e.g., via tokenization of time series data, but this is not done.

**Audience:**

No

**Audience Explanation:**

This submission is not of interest to a subset of the TMLR audience, in my opinion. Its theoretical insights are simple and incomplete, and the experiments are extremely limited.

**Claims And Evidence:**

No

**Claims Explanation:**

The claims of the paper are not supported by the evidence. Please see the above points, "rigor" and "experiments".

**Requested Changes:**

I do not find it likely that changes short of a total rewrite would be sufficient for acceptance. This includes more substantial theorems, full proofs, more thorough experiments, and better references to contextualize the contribution.

---

> ### Author Response · Authors · 2026-05-31
> **Response to Reviewer CTtd**
>
> We thank the reviewer for the critical comments.
> We agree that the manuscript should better contextualize the work within recent machine-learning literature,
> especially time-series tokenization and symbolic representations.
> In a revision, we will expand the related work accordingly and clarify that
> the paper is not proposing a new tokenizer or a state-of-the-art forecasting architecture,
> but rather a controlled diagnostic study of how discretization affects observable predictive structure.
>
> We also agree that some claims should be stated more narrowly.
> We will revise the framing so that the main empirical claim is limited to the same-(K) HMM benchmark:
> fixed latent-state cardinality does not by itself determine observable Bayes-level context sensitivity,
> and learner-side recovery can be family-dependent and non-monotone in an architecture-specific width parameter.
>
> We will also strengthen the rigor and experimental scope.
> The rank statement will be presented as a modest structural proposition with proof,
> and we are adding a causal Transformer learner in addition to the GRU
> under the same candidate processes, data splits, training protocol, seeds, and metrics.

---

### Review · Reviewer_BTMJ · 2026-05-29

**Summary Of Contributions:**

The paper evaluates discretization of continuous time series into symbolic sequences and how it affects predictive accuracy using Hidden Markov Models (HMMs) with fixed hidden-state counts (same-K families). It argues that forecasting difficulty is not determined by latent-state cardinality alone, but by the predictive structure following discretization.

Even with identical hidden-state counts, HMM families differ substantially in context sensitivity. Larger models may show worse performance. The overall goal is to separate model size from effective predictive difficulty.

**Additional Comments:**

n/a

**Audience:**

Yes

**Audience Explanation:**

Although the subject of discretization has been studied and evaluated extensively both in the context of time series but also in other areas of machine learning, it has now obtained increased relevance because of large language models.

I believe that the analysis methodology used here to evaluate discretization and generalization w.r.t. scale is significant. It is not necessarily novel and the authors acknowledge that the individual components are well-established, not novel. The claim of novelty may be in putting it all together: applying Hankel-rank to evaluate what discretization does to forecasting difficulty and how symbolic coarsening may collapse.

As such, the experimental design is the main contribution: constructing synthetic HMM families that deliberately hold latent-state cardinality fixed while varying observable context sensitivity. It may be argued that the theoretical results are unsurprising.

To me the most important contribution would therefore be the experimental methodology, in case it can be applied systematically by others, particularly in the analysis of LLM scaling and how HMMs have been used in practice to constrain token generation.

**Claims And Evidence:**

Yes

**Claims Explanation:**

The evaluation is limited to synthetic HMMs with K=4 and a single GRU architecture, leaving open whether findings generalize to real-world data or other sequence models (transformers). It is unclear with this limited empirical study whether observed differences in results are practically meaningful or partly noise-driven. The non-monotonicity result also seems to be based on a very small training set, making it difficult to judge the impact of potential overfitting.

Otherwise, the experiments do take into account the impact of initial conditions, training epochs/sample size and use multiple metrics, but use of only K=4 with a small latent state grid is limited and without statistical significance results being reported for comparison.

**Requested Changes:**

Additional experimental results are needed.

Is there a specific reason why the time series needs to be stationary?

Why use only GRU predictors?

---

> ### Author Response · Authors · 2026-05-31
> **Response to Reviewer BTMJ**
>
> We thank the reviewer for the constructive comments and for recognizing the value of the same-(K) diagnostic design. We agree that the empirical scope and the role of stationarity should be clarified.
>
> In a revision, we will strengthen the experimental section by adding a causal Transformer predictor in addition to the GRU, evaluated under the same candidate HMM processes, data splits, training protocol, width sweep, seeds, and metrics. This is intended to address whether the observed recovery trends are specific to a recurrent learner.
>
> We will also clarify why stationarity is assumed. Stationarity is not essential to the broad motivation of the work; rather, it allows us to define a time-invariant Bayes context-response curve and a single block Hankel matrix without explicit time indices. We will state this more clearly and include it among the limitations of the diagnostic setup.
>
> Finally, we agree that statistical uncertainty should be reported more explicitly. We will clarify the number of candidate-process/seed runs used for each point, explain the error bars in the figures, and avoid interpreting the observed non-monotonicity as a universal overfitting claim. Instead, we will present it as controlled diagnostic evidence within the same-(K) benchmark.

---

### Decision · Action_Editor_SBXA · 2026-06-17

**Recommendation:** Reject

**Audience:**

No

**Audience Explanation:**

The opinions here were a bit mixed among the referees. While the topic addressed by the paper is in principle important and interesting, the lack of sufficient empirical and theoretical support, and essentially ignoring ~10 years of literature, lead this contribution to have little practical relevance for the current discussion.

**Claims And Evidence:**

No

**Claims Explanation:**

All referees brought up a number of serious limitations with this study, including

- very limited experimental setup (only K=4 HMMs)
- very limited suite of models tested (only GRU)
- lack of statistical evaluation
- lack of nontrivial math. results and/ or full proofs
- neglect of recent literature

I concur with the referees' views.